# The Role of RNA-Binding Proteins in Hematological Malignancies

**DOI:** 10.3390/ijms23179552

**Published:** 2022-08-23

**Authors:** Pedro Aguilar-Garrido, Álvaro Otero-Sobrino, Miguel Ángel Navarro-Aguadero, María Velasco-Estévez, Miguel Gallardo

**Affiliations:** 1H12O-CNIO Hematological Malignancies Clinical Research Unit, CNIO, 28029 Madrid, Spain; 2Department of Hematology, Hospital Universitario 12 de Octubre, Instituto de Investigación Sanitaria Hospital 12 de Octubre (imas12), 28041 Madrid, Spain

**Keywords:** RNA-binding proteins, leukemia, lymphoma, myeloma, dysplasia

## Abstract

Hematological malignancies comprise a plethora of different neoplasms, such as leukemia, lymphoma, and myeloma, plus a myriad of dysplasia, such as myelodysplastic syndromes or anemias. Despite all the advances in patient care and the development of new therapies, some of these malignancies remain incurable, mainly due to resistance and refractoriness to treatment. Therefore, there is an unmet clinical need to identify new biomarkers and potential therapeutic targets that play a role in treatment resistance and contribute to the poor outcomes of these tumors. RNA-binding proteins (RBPs) are a diverse class of proteins that interact with transcripts and noncoding RNAs and are involved in every step of the post-transcriptional processing of transcripts. Dysregulation of RBPs has been associated with the development of hematological malignancies, making them potential valuable biomarkers and potential therapeutic targets. Although a number of dysregulated RBPs have been identified in hematological malignancies, there is a critical need to understand the biology underlying their contribution to pathology, such as the spatiotemporal context and molecular mechanisms involved. In this review, we emphasize the importance of deciphering the regulatory mechanisms of RBPs to pinpoint novel therapeutic targets that could drive or contribute to hematological malignancy biology.

## 1. Introduction

Hematological malignancies comprise a heterogeneous group of cancers that includes diverse and biologically different subgroups of neoplasms. The proposed WHO classification of hematological malignancies stratifies these neoplasms according to their original cell lineage: myeloid neoplasms, lymphoid neoplasms, mast cell disorders, and histiocytic neoplasms [1]. Each type varies in incidence, etiology, prognosis, and survival, and they are all essential contributors to the global cancer burden. In 2020, hematological neoplasms accounted for more than 7% of all diagnosed cancers worldwide and this number rose to 10% in the United States of America (USA) [2]. The incidence of hematological neoplasms varies depending on subtype, age, sex, and socioeconomic state of the patient [3]. In the USA alone, circa 200,000 people get diagnosed annually with hematological malignancies, the most common one being non-Hodgkin lymphoma followed by leukemia and multiple myeloma [2,4], and they account for 9.7% of all cancer deaths [2]. 

Relative survival for many hematological neoplasms has significantly increased in the past decade. Most hematological neoplasms have had the same robust frontline treatment for the past decades, such as Rituximab to treat lymphoma (approved in 1997), Bortezomib and Lenalidomide to treat myeloma (approved in 2003 and 2005, respectively), or Imatinib for chronic myeloid leukemia (CML) (approved in 2005). Indeed, there is a 10-year relative survival greater than 50% for most hematological malignancies, except for acute leukemia and multiple myeloma [5]. However, in the past 5 years, the emergence of novel therapies targeting poor prognosis neoplasms and/or relapse subset of patients is starting to change the clinical situation. Currently, these malignancies can be treated with multiple treatment regimens and therapy combinations, including precision medicine such as target therapy molecules (e.g., small molecule inhibitors), monoclonal antibodies, antibody-drug conjugates (ADC), bispecific T-cell engagers (BiTEs), immunotoxins, immune checkpoint inhibitors and Chimeric Antigen Receptor cell technology, efficiently improving the survival of these patients [6,7,8,9,10,11].

However, although there has been an improvement in the development of new therapies and medical care of patients, the heterogeneity of hematological malignancies constitutes a therapeutic obstacle, and the current small molecule inhibitors available in the clinic only target a subset of patients. Tumor cells have various strategies to evade/escape therapy, such as gain of genetic mutations, epigenetic alterations, abnormal drug metabolism, signaling pathway dysfunction, tumor microenvironment rewiring and stem cell persistence [12,13,14]. This way, malignant cells circumvent current therapies and survive, leading to therapy resistance, relapse, and failure of treatments.

Therefore, there is an urgent unmet clinical need to identify new biomarkers and potential therapeutic targets that would contribute to overcoming the current therapy resistance and poor outcomes of hematological neoplasms. Here, we review the role and potential clinical significance of RNA-binding proteins (RBPs) in hematological malignancies, both as biomarkers and therapeutic targets for different neoplasms.

## 2. RNA-Binding Proteins

Cancer involves a myriad of cellular and molecular mechanisms; however, transcription and translation have gained more interest in cancer research as they are critical in gene expression and protein biogenesis and seem to highly contribute to the development and progression of a number of different tumors. Nevertheless, the underlying mechanisms causing these alterations remain unknown.

Gene transcription and mRNA translation are tightly regulated processes where RBPs are closely involved. These RBPs represent a group of regulatory proteins with structural domains with high affinity for specific RNA sequences, which are key components of RNA biogenesis [15]. RBPs are increasingly becoming more popular as they are essential for the proper formation of blood cellular components (termed hematopoiesis) [16]. Indeed, studies of gene expression profiling have demonstrated that different blood cell differentiation stages are not only regulated by transcription factors but also influenced by post-transcriptional regulatory mechanisms, such as those controlled by RBPs [17,18]. For this reason, RBPs can be key players in the onset and progression of hematological malignancies.

RBPs are involved in many aspects of the post-transcriptional modulation of mRNAs and are deeply regulated by post-translational modifications (PTMs), such as phosphorylation, acetylation, or methylation. Depending on the PTMs acquired and the balance between them, the properties of RBPs dramatically change, affecting their interaction with other molecules, their biological function, and their subcellular localization [19]. Because RBPs are implicated in multiple cellular processes, perturbations in RBP–RNA complex activity have been associated with cancer development and progression. In recent years, an increasing number of studies have pointed out the key role of RBPs in hematological malignancies [20,21,22].

To date, more than 500 RBPs have been characterized in humans, all of which have different affinities for RNA [23]. Although they can also interact with proteins and DNA, most of the identified RBPs specifically bind to RNAs (including mRNA) through the recognition of specific sequences mostly located in the untranslated regions (UTRs) [24,25]. Within the structure of most known RBPs, there is a small group of “classic” specific domains known as RNA-binding domains (RBDs) which mediate the interaction between RBPs and RNAs in a sequence- and structure-dependent manner. These include the zinc-finger and DEAD/DEAH box helicase, RNA recognition motif (RRM), or K homology (KH), among others (Table 1). These domains usually contain between 60–100 amino acids that normally adopt an αβ topology [26]. 

Commonly, RBPs contain multiple repeats of the same RBD, which contributes to an enhanced specificity and affinity for their target mRNAs [27,54]. Nonetheless, other RBPs are also able to interact with mRNA through “nonclassic” RBDs, thus allowing them to bind to a wider spectrum of molecules [24].

RBPs and transcripts assemble in what are termed ribonucleocomplexes to control gene expression. These dynamic interactions determine the fate of the transcripts in every step of its cycle, from synthesis to decay, depending on the combination and balance of each type of RBP forming the ribonucleocomplex at each time point [59,60]. Because the same recognition sequences could be shared by many different RNAs, the expression of multiple mRNAs could be controlled by a single RBP. Therefore, each RBP can regulate a plethora of molecular pathways and different physiological processes depending on which mRNAs are expressed [61].

Despite the complexity of the genomic and mRNA network, this system has great versatility due to the continuous modification of the transcriptome as a result of the high activity rate of the mRNA machinery. Depending on the cellular needs at a given time, the gene regulatory networks persistently adjust mRNA and protein levels, influencing from transcription to protein degradation, which includes mRNA transport and maturation and translation [62,63,64].

## 3. mRNA Regulation by RBPs

The information coded in the DNA must be transcribed by RNA polymerases into RNA molecules to generate proteins. These primary transcripts are usually modified and processed to mature them into functional molecules. Therefore, the extensive processing of protein-encoding pre-mRNA transcripts is essential in eukaryotic gene expression [65]. As mentioned before, RBPs are involved in many steps of RNA metabolism and have the ability to control post-transcriptional gene regulation by many mechanisms.

### 3.1. mRNA 5′ Capping

The production of mRNA requires the synthesis of a pre-mRNA followed by a 5′ capping, deletion of introns by splicing, and a final 3′ cleavage and polyadenylation, in order to produce a mature mRNA. The addition of a 7-methylguanosine to the first nucleotide of the 5′ of a pre-mRNA through a triphosphate bound is known as capping [66]. Just before finishing transcription, pre-mRNAs are capped. This process is essential for the growth of eukaryotic cells, as it participates in many functions such as the regulation of nuclear export, the prevention of mRNA degradation, the fostering of translation and the promotion of 5′ proximal intron excision [66,67]. This 5′ capping is a tightly regulated process that allows the generation of a stable and mature mRNA molecule able to undergo translation by the ribosome.

In mammals, all 5′ caps of pre-mRNAs bind to a heterodimer known as the nuclear cap binding complex (CBC), which is formed by two different proteins: CBP20 (cap binding protein 20) and CBP80 (cap binding protein 80). This heterodimer protein complex participates in monitoring mRNA quality through the nonsense-mediated mRNA decay pathway (NMD), in which there is an elimination of mRNA transcripts that contain premature stop codons, leading to a reduction in errors in gene expression [68].

Prevention of mRNA degradation is one of the principal functions of mRNA capping. It prevents the digestion of mRNAs by the exoribonuclease activity of Xrn1 and Xrn2, to continue its journey to the ribosome. Once in the cytoplasm, 5′ capping acts as a trigger for the recruitment of the translational machinery to the 5′ end of protein-encoding RNAs, inducing the recognition of the cap by translation initiation factors (e.g., eIF4E) and leading to the recruitment of the 40S ribosome subunit to the 5′ end [69,70].

### 3.2. Splicing of Pre-mRNAs

The process of removing introns (noncoding regions of RNA) and splicing exons (coding regions) back together from a pre-mRNA to generate a mature mRNA is known as RNA splicing, and it is crucial for bona fide protein expression.

In eukaryotes, splicing mainly occurs in a series of reactions catalyzed by the spliceosome, a huge macromolecular complex consisting of small nuclear ribonucleoproteins (snRNPs) together with more than 100 different other proteins. However, most transcripts can undergo alternative splicing as well, in which a selection of alternative exons could be executed, resulting in the production of different isoforms of the same mRNA [71].

Due to this alternative splicing, different mRNA isoforms are produced in different tissues and cell types within them. This process alters gene coding and has an impact on mRNA stability. It has been recognized as a mechanism to increase the functional diversity of the proteome, leading to an additional layer of complexity for gene expression regulation. Splicing processes can occur both during transcription of mRNAs (cotranscriptional) or after this process (post-transcriptional) [72,73].

Many proteins of the spliceosome are well-studied RBPs, such as DExD/H type RNA-dependent ATPase. These are RBPs with helicase activity and they are implicated in rearrangements, single-strand RNA translocation, strand annealing and protein displacement. Due to the RBPs’ wide range of molecular processes implication, a plethora of diseases are linked to mutations in many RBPs involved in splicing and splicing regulation [26,74].

### 3.3. Cleavage and 3′ End Formation

The processes of cleavage and 3′ end formation are fundamental steps in the maturation of pre-mRNAs into functional mRNAs that can be exported from the nucleus to the cytosol.

The final stage of transcription is marked by endonucleolytic cleavage, prior to 3′ end formation, where the 3′ end of a newly pre-mRNA is first cleaved-off by a set of proteins, followed by the addition of a poly(A) tail at the 3′ end of an mRNA (polyadenylation) of approximately 200 adenosine residues [75].

The poly(A) tail is important for nuclear export to the cytosol, translation initiation and stability of mRNA. Similar to splicing, in some genes, transcripts can be polyadenylated at several possible sites, leading to the generation of more than one transcript from a single gene. This process is known as alternative polyadenylation and alters the stability, localization and transport of these pre-mRNAs. Within the human genome, it is estimated that half of the genes, at least, are subject to this alternative splicing and polyadenylation, generating isoforms that differ in the 3′ UTR length or, even, encoding different proteins [76].

The polyadenylation machinery consists of six multimeric proteins that act together to, first, mediate the cleavage of newly nascent pre-mRNA and, second, allow the polyadenylation of the transcript. It is as assembled by a cleavage stimulatory factor (CstF), cleavage and polyadenylation specificity factor (CPSF), cleavage factors I and II (CFI and CFII), poly(A) polymerase (PAP) and poly(A) binding protein II (PABP II) [75,76].

### 3.4. mRNA Export

The largest cell organelle is the nucleus, surrounded by the nuclear envelope, which is pierced by nuclear pore complexes (NPCs). These are the main gateways through which mRNAs are transported between the nucleus and the cytosol, although this translocation requires specific transport receptors to break this barrier.

The mRNA export machinery includes numerous RBPs (e.g., hnRNP C tetramer), adaptor proteins (EJC and SR) and proteins that associate with NPCs (TREX complex). All of these proteins interact with the mRNA at some point, facilitating the transport from the nucleus to the cytosol [77,78]. After mRNA is exported from the nucleus, all the interacting RBPs can either remain in the nucleus or follow the transcript into the cytosol. Once in this compartment, mRNAs could remain interacting with the same RBPs or they can be recruited by others, which determines mRNA cytosol compartment localization [79].

### 3.5. mRNA Stability

The abundance of an mRNA depends not only on its synthesis, processing and nuclear export rates, but also on its degradation rate in the cytosol. Thus, mRNA stability maintenance plays a key role in the translation of essential proteins and in the degradation of transcripts mediated by the proteasome.

Cytosolic stability of mRNAs relies upon mRNA interactions with RBPs and other types of proteins, which are crucial for the modulation of this stability. This process can be quickly modulated to alter specific gene expression, giving a huge flexibility in affecting changes in patterns of protein synthesis [80].

RBPs contribute to controlling the degradation of transcripts, which is an important step and is highly variable. One of the main processes observed in the alteration of mRNA stability is the alteration of the length of the poly(A) tail, which is related to the earlier decapping [81].

### 3.6. Translation

The translation process is one of the most well-studied and best-detailed processes in which RBPs play a key role. Briefly, the regulation of the translation process is controlled by three main mechanisms: poly(A) tail modification, interaction and association with RBPs, and assembly of the translation machinery [82].

The increase in the poly(A) tail length of an mRNA can significantly improve translation rates, which depends on the activity of PABP, an RBP responsible for recruiting many factors essential for the initiation of translation [83].

Transcript-specific cytosolic compartmentalization of RBPs constitutes a targeted location for processing, sorting, storage or degradation. Some RBPs can be associated with exonucleases to be degraded, while others can be protected and remain silent until they are needed in the ribosome for protein production [84].

### 3.7. RNA Editing

The process of insertion, deletion or subtraction of nucleotides in RNA sequences is known as RNA editing. The most common RNA editing process is the exchange of adenosine to inosine (A-I), catalyzed by the RBP family ADARs, such ADAR1, but with the regulation and collaboration of other RBPs such as DROSHA or TARDBP [85]. A-I editing promotes deep modification of transcriptomic landscape by protein modification, RNA splicing, export or microRNAs activity.

## 4. RNA-Binding Proteins in Hematological Malignancies

Diseases such as neurodegeneration, dysplasia, cancer, ribosomopathies and hematological malignancies can all be caused by defects in the normal functionality or expression of RBPs. Currently, RBPs are understood to be crucial for healthy hematopoiesis and to play a significant role in hematological neoplasms by functioning as both oncogenes and tumor suppressors [86,87,88,89].

RBP modifications result in biological changes such as altered hematopoietic lineage development or abnormalities that, in many circumstances, can result in bone marrow failure [90,91]. Likewise, alterations in several RBPs also cause variations in splicing in myeloid and lymphoid cell lineages [16]. Because RBPs act as nodes for multiple signaling pathways and take part in hematological malignancies, they offer a therapeutic opportunity in the clinic, as potential new biomarkers as well as new therapeutic targets for hematological neoplasms Figure 1.

Some of the most significant RBPs, whose dysregulation has been associated with the development of hematological malignancies, can be found summarized in Table 2.

### 4.1. RBM39–mRNA Splicing

RNA-binding motif protein 39 (RBM39), also known as HCC1, CAPER or CAPERɑ, is a serine/arginine-rich RNA-binding protein implicated in the regulation of gene expression at several levels, such as transcription, alternative splicing, and translation [118]. RBM39 contains two central RRMs, an N-terminal arginine- and serine-rich domain (RS domain) which can be phosphorylated, and a C-terminal U2AF homology motif (UHM). Both the N- and C-terminal domains enable RBM39 to interact with other proteins. Thus, RBM39 plays a critical role in the activation of transcription factors, such as activating protein-1 (AP-1)/Jun, steroid receptors and transcriptional coactivators, such as ASC-2 [119,120]. Furthermore, c-Abl phosphorylates RBM39, enhancing its transcriptional activity [121]. RBM39 has been described as a potential splicing factor that acts in pre-mRNA alternative splicing by binding to RNA or recruiting specific splicing factors to regulate this process [113]. RBM39 is implicated in the recognition of 3′ splice sites and can interact with several molecules in early spliceosome assembly that contain the U2AF ligand motif (ULM), such as U2AF65, SF3B155, and RSRC1.

RBM39 is involved in several solid cancers, such as breast and lung cancer [114,122,123], but it is also implicated in hematological malignancies. Wang et al. [124] used a comprehensive screen to pinpoint RBM39 as a crucial RBP necessary for AML survival, and demonstrated therapeutic effects by pharmacologically reducing RBM39 protein in spliceosome mutant AML models. They identified RBM39 as a key AML-specific RBP, involved in AML malignant cell growth and maintenance [124]. Moreover, its depletion leads to a decrease in leukemia progression and an increase in overall survival. Likewise, it has been shown that RBM39 plays an important role in multiple myeloma, by promoting multiple myeloma cell proliferation and inhibiting apoptosis under hypoxic conditions in vitro [125]. Tong and collaborators also described that a long noncoding RNA, DARS-AS1, inhibits RBM39 degradation under hypoxic conditions. They demonstrated that DARS-AS1 is regulated by HIF-1α and that DARS-AS1 regulates mTOR signaling via RBM39, thus concluding that the HIF1α/DARS-AS1/RBM39 axis could be a useful target in multiple myeloma.

### 4.2. Musashi Proteins (MSI1 and MSI2)-mRNA Translation

The Musashi protein family is a group of RNA-binding proteins that play necessary roles in controlling stem and progenitor cell functions [109,110,126,127,128]. This family contains Musashi-1 (MSI1) and Musashi-2 (MSI2), two highly conserved homologous proteins [109]. MSI2 is transcribed ubiquitously, in contrast with MSI1, which is mainly found in neural stem cells or embryonic progenitor cells and is involved in neural stem cell proliferation and cell cycle regulation [128]. On the other hand, MSI2 is implicated in the self-renewal and differentiation of embryonic [129], neuronal [110] and hematopoietic stem cells [111]. This suggests that each member of this family is regulated in a context-dependent manner.

The Musashi RNA-binding proteins constitute a subgroup within the heterogeneous nuclear ribonucleoprotein (hnRNP) A/B type of proteins that contain two N-terminal copies of RNA recognition motifs (RRMs) in addition to a C-terminal PABP-interacting domain [109]. RRM1 mainly binds to RNA whereas RRM2 has a complementary role, stabilizing the protein–RNA complex and increasing RNA binding affinity in combination with RRM1 [41]. MSI1 and MSI2 recognize different mRNA motifs, preferentially within the 3′-UTR (3′ untranslated region) of target transcripts, although it is thought that they regulate similar mRNA targets as there is a partial overlap between MSI1 and MSI2 binding targets [130]. The C-terminal region of MSI proteins allows protein–protein interactions, containing several protein-binding domains and contributing to the stimulation or repression of protein translation. In contrast with MSI1, which only has one isoform, the MSI2 gene produces four different isoforms (MSI2a, MSI2b, MSI2c and MSI2d), all of them containing the same RRM motifs and PABP domain, but differing on the C-ter and N-ter amino acids, which makes them susceptible to different PTMs [131]. The canonical MSI2a can be phosphorylated at its C-terminal, converting it from a translational repressor to an antivator, while MSI2b lacks this regulatory site and thus cannot promote translation of target mRNAs [132]. Therefore, MSI2 can lead to the translation or repression of different target mRNAs, and despite the exact underlying molecular mechanisms being not yet fully clear, it seems that the regulation by MSI2 depends on the isoform, phosphorylation status, and cellular context [131,132].

Elevated expression of Musashi proteins has been identified in several solid tumors [132] and this overexpression is extensively correlated with cancer cell proliferation. In contrast with MSI1, MSI2 expression has been predominantly reported in hematological neoplasms, fundamentally in leukemia, probably due to the fact that MSI2 is mainly expressed in the hematopoietic system and is an important regulator of hematopoietic stem cells (HSCs) [111]. High MSI2 expression levels are found in nearly all hematological malignancies and are associated with poor prognosis [130,133].

Functional studies have shown an essential role of MSI2 in leukemic progression and stem cell renewal. Kharas et al. described that MSI2 dysregulation is associated with aggressive AML cases and that its inhibition in human AML cell lines resulted in reduced proliferation and increased apoptosis [111]. HoxA9, c-Myc and Ikzf2 were shown to be targets of MSI2, maintaining the oncogenic leukemia stem cell (LSC) self-renewal program in AML [134]. The most prominent cancer pathway regulated by Musashi proteins is the Notch pathway. The main studied target of Musashi proteins is Numb, an inhibitor of the Notch signaling pathway [135]. The Musashi2-Numb axis is crucial to regulate chronic myeloid leukemia (CML) progression to blast crisis, and its depletion results in reduced leukemia development in CML models [136]. In chronic lymphocytic leukemia (CLL), MSI2 is implicated in CLL cell growth and survival, and correlates with worse clinical outcome [137]. Palacios and colleagues described critical functions of MSI2 in CLL progression, probing its driving role in proliferative CLL cells. They also demonstrated that Ro 08-2750, an MSI2 small molecule inhibitor whose efficacy was previously reported in a murine AML model [138], preferentially and more effectively targeted leukemic B cells and myeloid cells while not affecting HSCs [105]. MSI2 is also involved in myelodysplastic syndrome (MDS) progression, as conditional deletion of MSI2 in mouse models led to a reversal of MDS phenotypes, and its overexpression resulted in MDS progression [105].

### 4.3. IGF2BP3–mRNA Localization, mRNA Stability and mRNA Degradation

Insulin-like growth factor 2 mRNA binding protein 3 (IGF2BP3), also known as IMP3, KOC, CT98, KOC1, and VICKZ3, belongs to the IGF2BP family, which is composed of three highly structurally and functionally related proteins. These RBPs carry two amino-terminal RRMs and four carboxy-terminal KH domains [139]. Although several studies report that these proteins interact with RNA via KH domains [140,141], it has been subsequently described that RRMs also contribute to RNA binding, enabling a very specific interaction with RNA targets [142]. IGF2BP3 expression follows an oncofetal pattern, since it is highly expressed during embryonic development [139], while its levels decrease at the late stages of embryogenesis and it is upregulated in several adult cancers [143,144].

IGF2BP3 controls the localization, stability and degradation of its mRNA targets [143]. This protein is primarily cytoplasmic, but it can also act in the nucleus where it binds target transcripts and facilitates their nuclear export [145] due to the presence of nuclear export signals within the KH2 and KH4 domains [139]. IGF2BP3 has the bimodal capacity to regulate mRNA fate, avoiding or promoting binding to the RNA-induced silencing complex (RISC). Functional studies reveal that IGF2BP3 directly interacts with ribonuclease XRN2 [146] and deubiquitinase USP10 [147], leading to EIF4EBP mRNA degradation or tumor suppressor p53 reduction, respectively. It has been observed that IGF2BP3 is highly expressed in the majority of solid cancers, promoting a proliferative phenotype, metastatic activity and poor outcomes of these tumors [106].

This RBP has been identified as a key player in many hematological malignancies as well. IGF2BP3 is necessary for B-cell acute lymphoblastic leukemia (B-ALL) cell survival. Its overexpression in the bone marrow of mice causes abnormal growth of hematopoietic stem cells [107]. In 2021, Tran et al. demonstrated the critical role of IGF2BP3 in mixed-lineage leukemia (MLL)-rearranged leukemia, a subtype of acute leukemia with poor outcomes, high relapse and therapy refractoriness. They proved that IGF2BP3 itself is a direct target of MLL-AF4 and regulates a post-transcriptional operon that increases the expression of leukemogenic genes [148]. A recent study by Mäkinen et al. also showed that IGF2BP3 is associated with active cellular proliferation in B-cell blasts [108]. Surprisingly, they found that high IGF2BP3 mRNA levels were related to improved survival in a high-risk pediatric B-ALL cohort.

In multiple myeloma, IGF2BP3 expression is inhibited by miR-9-5p, which is upregulated after AR-42 treatment, a histone deacetylase inhibitor [149]. IGF2BP3 controls CD44, a glycoprotein that has been associated with current therapy resistance in multiple myeloma.

Related to myelodysplastic syndromes, apoptotic bodies from cytosine arabinose-resistant cells notably contain IGF2BP3, which promotes the survival of recipient cells by activating the PI3K-Akt and p42-44 MAPK pathways via c-Myc [150].

### 4.4. hnRNP K–mRNA Localization, Transcription, Translation and Splicing

Heterogeneous nuclear ribonucleoprotein K (hnRNP K) is a multifunctional RBP implicated in a myriad of biological processes, such as transcription, translation, splicing, RNA stability and chromatin remodeling, among others [29,95]. It contains three KH domains (KH1, KH2 and KH3) responsible for nucleic acid binding [151], one K protein interactive domain (KI) responsible for protein binding, and a nuclear-cytoplasmic shuttling domain (KNS) that confers the ability to translocate bidirectionally through the nuclear pore complex [152].

hnRNP K is an intracellular protein that localizes both in the nucleus and the cytoplasm and is ubiquitously expressed in all tissues. This protein could be modified by many other proteins, as it can be methylated, acetylated, sumoylated or phosphorylated. Depending on this PTM balance, hnRNP K can regulate its interactions with different molecules and influence its functions, having a huge impact on many biological pathways [95]. Because of its multiple interactions with many proteins, alterations of its protein levels can be oncogenic by deregulating the transcription and/or translation of multiple oncogenes or tumor suppressors, supporting the hypothesis that RBPs are involved in tumorigenesis [96].

Many studies have demonstrated the link between hnRNP K deregulation and cancer. hnRNP K is a key player in DNA damage response as it is required for p53-mediated transcription of cell cycle checkpoint genes [97,153], and thus could promote cell proliferation via p21, a negative regulator of cyclin-dependent kinases (CDKs), and its subsequent reduction in G0/G1 stage cells [154]. Numerous studies have shown that long noncoding RNAs (lncRNAs) influence the growth of tumors by either promoting or inhibiting hnRNP K expression [98,154,155,156,157].

It has been reported that approximately 2% of AML patients have a 9q21.32 deletion that encompasses the *HNRNPK* gene [99,100,101,158]. For the first time, Gallardo et al. described that this haploinsufficiency is linked to the onset of AML phenotypes in animal models, pointing out hnRNP K’s tumor suppressor role [21]. Moreover, years later, Naarman-de Vries IS et al. found that the 9q21.32 deletion produces a reduction in hnRNP K-targeted gene expression, such as *CDKN1A* and *CEBPA*, which is related to AML del(9q) pathogenesis. Furthermore, it seems that hnRNP K influences the differentiation of AML-derived cells through its interaction with the transcription factor PU.1 [102].

Related to leukemia, several studies have demonstrated that hnRNP K also has an impact in chronic myeloid leukemia (CML) [103,104]. The activity of BCR/ABL kinase induces hnRNP K expression by enhancing hnRNP K gene transcription and mRNA stability. As mentioned before, c-Myc mRNA can be overexpressed due to hnRNP K overexpression, a situation that is essential for the phenotype of CML cell progenitors.

Recently, Gallardo et al. found that diffuse large B-cell lymphoma (DLBCL) patients with high hnRNP K expression suffered poor clinical outcomes compared with patients with low hnRNP K expression [22]. In contrast with previous studies [21], they showed that hnRNP K could act as an oncogene forcing c-Myc expression and contributing to DLBCL pathogenesis.

Multiple myeloma (MM) pathogenesis is also correlated with hnRNP K deregulation. The study of Evans JR et al. showed that at least 42% of MM patients carry a mutated version of the c-Myc IRES that binds more tightly to hnRNP K. Thus, the upregulation of hnRNP K leads to an increase in c-Myc, contributing to MM development [159].

Moving out of neoplasms, the presence of hnRNP K autoantibodies has been found in the serum of patients with immune-mediated aplastic aplasia (AA), a type of bone marrow failure syndrome. Qi Z et al. suggested that specific immune responses to hnRNP K may induce the polarization of Th1 CD4+ cells and may contribute to the development of AA. It seems that the destruction of immature hematopoietic cells overexpressing hnRNP K may induce a specific immune response to this RBP in patients with immune-mediated AA [90].

Lastly, Aguilar-Garrido et al. described the role of hnRNP K in non-neoplastic hematological malignancies. In vivo overexpression of hnRNP K promoted the exhaustion of hematopoietic stem cells by ribosome dysfunction, inducing a bone marrow failure phenotype [91].

## 5. Discussion

The fate of mRNAs mostly depends on their relationship and network with RBPs, but the principles underlying these interactions remain poorly understood. How RBPs achieve binding specificity through their different RBDs and how they interact and compete with other RBPs is something not yet fully characterized [160]. Furthermore, the precise mRNA sequences that RBPs bind are still being fully identified, and it is unknown whether they are cell context specific.

RBPs are crucial for controlling gene expression at every stage, and their dysregulation can result in a wide variety of hematopoietic malignancies [88]. Excitement and rising interest in this new topic have been generated by our growing understanding of the discovery of these unregulated RBPs and their function in hematological malignancies.

However, the list of unconventional RBPs with newly discovered biological roles continues to grow, as is the case for RBM39, MSI1, MSI2, IGFBP3 and hnRNP K.

Strikingly, several RBPs have both oncogenic and tumor suppressor functions, behaving or regulating both oncogenes and tumor suppressor genes [161]. Which of these two paths is chosen depends on the spatial–temporal context: the type of cell/tissue, the stage the cell is at and the microenvironment surrounding the tumor [162].

Now, we are becoming aware of the complex control enforced after the point of transcription. RBPs play a significant role in this regulation, which allows coordinated remodeling of the transcriptome and proteome in response to microenvironmental influences.

It has long been assumed that a protein’s activity is mostly regulated and influenced by interactions with other proteins: however, it is also possible that this situation could also be influenced by its interaction with RNA, in the case of RBPs.

Additionally, developments in sequencing technology and novel techniques to decipher the RBP landscape are improving our understanding of the key players involved in gene expression, translation and post-transcriptional regulation mechanisms. Genetic, post-transcriptional or post-translational modifications to a single RBP may have an impact on a variety of RNA targets but also at multiple points of their RNA metabolism. Thus, analyzing these protein–RNA structural interactions will be enlightening.

New approaches for therapeutic targeting are now emerging from a better understanding of how RBPs affect this regulatory network in healthy and malignant hematopoiesis. These strategies may focus on the RBP itself, its RNA interaction, the up/downstream alterations to the proteome brought on by changes in RBP function, or any combination of these possibilities. For instance, new therapeutic strategies target the RBPs themselves in hematological malignancies, such as the use of XPO1 inhibitors for both myeloma (e.g., STORM clinical trial, ID: NCT02336815) and lymphoma (ID: NCT05422066) [163], MSI2 inhibitors for AML (ID: NCT01546038) [138], or SRSF2 inhibitors for MDS [164]; but also the inhibition of their interaction and/or downregulation of their targets comprise a new therapeutic horizon, such as the targeting of HNRNPK and c-MYC [22].

Although there is an increasing development of new drugs targeting RBPs and new therapeutic strategies are being discovered, there are still many questions that need to be answered. There are multiple dilemmas regarding the use of RBPs inhibitors. For instance, many RBPs act as master regulators in cancer, behaving as both oncogenes and tumor suppressors; most RBPs have multiple locations in the cell related with their function and a promiscuous nature with multiple targets and functions. All of these can lead to unspecificity, side effects and toxicity. Thus, we need to understand the physiological implications of altered RBPs, which type of complexes they form and their dynamics, what is the PTM’s role on altered RBPs and what kind of structural information is needed to target RBPs (isolated RBP versus RBP-RNA complex). Nevertheless, these are similar disadvantages to the traditional chemotherapy, which is still very effective and kept as the current frontline treatment for many cancers.

## 6. Conclusions

In conclusion, we highlight how crucial it is to comprehend the regulation of RBPs, which are emerging biomarkers of hematological malignancies that behave as master switches in cancer, aging and hematopoiesis, in order to identify new targets and pathways that drive or contribute to hematological diseases, neoplasms and dysplasia. In this context, future research will be essential in identifying relevant RBPs, their targets and their working network.

## Figures and Tables

**Figure 1 ijms-23-09552-f001:**
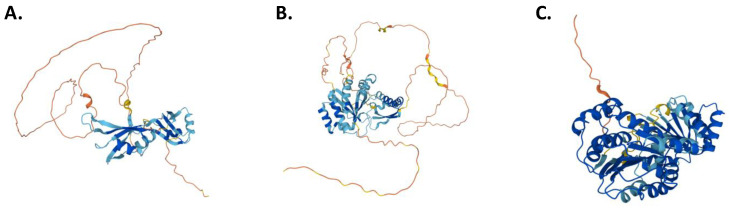
Structure of (**A**) MSI2, (**B**) hnRNP K, and (**C**) EIF4A1 proteins. Structure schema obtained from AlphaFold (AF-Q96DH6, AF-P61979, and AF-P60842, respectively) (https://alphafold.ebi.ac.uk/assets/License-Disclaimer.pdf, accessed on 7 August 2022, “not changes were made”) [92,93].

**Table 1 ijms-23-09552-t001:** RNA-binding domains classification.

Type of Binding Domain	Structure Interface	Interaction between RBP and RNA	Nucleic Acid Affinity	RBPs Containing RBDs
KH type I (hnRNP K homology type I)	Structure approximately 70 amino acids long. It typically adopts βααββα, forming by a β-sheet composed of 3 antiparallel β-strands and 3 α-helices [27,28]	Four single-stranded nucleotides are recognized by the invariant Gly–X–X–Gly motif, the near helices, and the β-strand that follows α2 (type I)	C-rich ssDNA and ssRNA [27,28]	hnRNP K [29], Nus A [30], SF1 [31], Nova-2 [32]
KH type II (hnRNP K homology type II)	Structure approximately 70 amino acids long. It is displayed as a αββααβ [27]	Same as KH I, where four single-stranded nucleotides are recognized by α3 (instead of α2 as in KH I)	C-rich ssDNA and ssRNA [27,28]	hnRNP K [29], Nus A [30], SF1 [31], Nova-2 [32]
KH type III (hnRNP K homology type III)	Structure approximately 75-80 amino acids long. It typically adopts a spatial configuration of βααββα, forming a 3-stranded β-sheet held against a 3-helix cluster [27]	Same as KH type I	C-rich ssDNA and ssRNA [27,28]	Nova-2 [33], hnRNP K [34]
RRMs (RNA recognition motifs)	Structure approximately 80-90 amino acids long. It typically adopts topology of βαββαβ, forming a 4-stranded β-sheet and 2 α-helices [35]	Main interaction between the binding domain and RNA is mediated through the β-sheet	Polypyrimidine (mainly C- and U-rich sequences) ssRNA [35,36]	U2AF65 [36], nucleolin [37], SRp20 [38], hnRNP F [39], FOX1 [40], Musashi 1, Musashi 2 [41], RBM39 [42]
ZnF (Zinc Fingers)	Structure approximately 30 amino acids long. It displays a ββα topology, forming a β-hairpin and an α-helix together with a Zn^2+^ ion [43]	Binding to nucleic acids through the α-helix	dsDNA, ssRNA, dsRNA [44]	MBNL1 [44], TFIIIA [45], ZRANB2 [46]
dsRBDs (double-stranded RNA binding domains)	Structure approximately 65-70 amino acids long. It typically adopts a αβββα topology, where 2 α-helices are packed along a 3-stranded anti-parallel β-sheet [47]	Binding to dsRNA backbone through α2 and the β1- β2 loop. Additional interactions occur through the α1	dsRNA [47]	ADAR1 [48], Dicer [49]
DEAD-box	Structure approximately 300-400 amino acids long. It adopts a βαββαβ topology, forming a 4-stranded β-sheet and 2 α-helices, similar to RRM binding domains [50]	Helicase core binds to the backbone of the RNA, without contact with the nucleotides	Polypyrimidine ssRNA [50]	eIF4A1/DDX2 [51], p68 [52], p72 [53]
PUF (Pumilio-fem-3 binding factor)	Structure approximately 6-8 tandem copies of a 35 amino acids long sequence. It adopts a topology of 3 α- helices, forming a triangle [54]	Binding to RNA is through the α 2 in each tandem repeat	ssRNA [55]	PUM1, PUM2 [55]
SAM (Sterile alpha motif)	Structure approximately 150-160 amino acids long. It displays a topology of 6 α-helices, packed by a hydrophobic core [56]	Traditionally known to bind protein, but has recently been shown to bind RNA	Hairpin RNA [56]	p63, p73 [57], p73, EPHA2 [58]

**Table 2 ijms-23-09552-t002:** Summary of RBPs implicated in hematological malignancies.

RBP	RNA Binding Motif	Role in Normal Hematopoiesis	Related Hematological Malignancies
ADAR1	dsRBD [48]	Regulation of HSCs differentiation via base editing activity	Enhanced editing activity in CML, increasing self-renewal capacity
DDX3X, DDX5	DEAD Box [51]	Essential for the innate immune response and normal hematopoiesis	Frequently mutated in hematological malignances, and upregulated upon Imatinib treatment
DDX21	DEAD Box [51]	HSCs self-renewal	Increases leukemia stem cell proliferation in AML
EIF4E	DEAD Box	Transcription factor, cell differentiation [94]	Blockage of myeloid differentiation, leading to leukemogenesis
hnRNP K	KH1, KH2, KH3 [95]	DNA damage response and cell cycle arrest [96,97]	Deletion in AML [98,99,100,101], overexpression in CML [102,103], oncogen in DLBCL [22] and MM [104]
IGF2BP3	RRM, KH [105]	Self-renewal of HSCs [106]	B-ALL cell survival [106], MLL [107], and therapeutic resistance in MM [108]
MSI2	RRM [41]	Self-renewal of HSCs [109]	Upregulated in most hematological malignancies and associated to poor prognosis [110,111]
RBM39	RRM [42]	Part of the spliceosome [112]	AML malignant cell growth and maintenance [113], as well as myeloma progression [114]
SRSF2	RRM [38]	Essential for myeloid hematopoiesis [115]	Mutations associated to poor survival in MDS [115]
SF3B1	RRM [36]	HSCs homeostasis [116]	Mutations associated with MDS [116]
ZFP36	ZnF [43]	Hematopoiesis and cell differentiation [117]	Loss of function leads to leukemogenesis [117]

## Data Availability

Not applicable.

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
