# Peer review of "The Role of RNA-Binding Proteins in Hematological Malignancies"

_ijms, 2022, doi:10.3390/ijms23179552_

Round 1
Reviewer 1 Report
This review summarized the role of RNA binding protein (RBP) in multiple steps of post-transcriptional gene regulation and RNA metabolism, highlighted the association of four RBPs (RBM39, MSI1/2, IGF2BP3, hnRNP K) with the development of hematological malignancies, and emphasized that the underlying biological mechanisms should be further explored so as to identify new targets and pathways that drive or contribute to hematological diseases. Before publication of this review, following suggestions should be considered:
Major points:
1. Since this is not the first review about RBPs’ role in hematological malignancies, an extensive summary of RBPs based on previous studies will be valuable. Suggest to summarize the RBPs, their targets, roles and related hematological malignancies in a table or a schematic diagram at the beginning of Section 4. RNA-binding proteins in hematological malignancies.
2. Although this review emphasized several times that RBPs are potential promising therapeutic targets for hematological malignancies, some features of RBPs including their subcellular location and complicated regulation network make draggability very challenging. It will add value to this review if such challenges and recent advances in novel approaches both in preclinical and clinical stages can be appropriately discussed in Section 5. Discussion.
3. For 4.2. Musashi proteins (MSI1 and MSI2) - mRNA translation, please focus on MSI2 and rephrase the description about MSI1 to make it more concise since MSI2 is a key player in hematological diseases in contrast to MSI1. Besides, the positive and negative regulation of MSI2 on mRNA translation in various settings and its relation with hematological malignancies were not sufficiently described here.
4. State the binding ability of RBD to dsRNA and ssRNA clearly in Table 1. Besides, it will improve the readability if a schematic diagram can be provided to vividly show the structure of binding domain of RBPs and the interaction between RBP and RNA.
5. In Section 3. mRNA regulation by RBPs, Is there any progress about RBP function in RNA editing?
6. In Section 1. Introduction, rephrase the current therapies and unmet medical needs to make it well-founded and accurate. For example, 10-year relative survival cannot reflect the most recent advances in novel therapies for hematological malignancies. Besides, NK-cell therapy has not been approved for any cancer, and the therapies listed in Paragraph 2 does not cover “small molecule inhibitors” which is mentioned in Paragraph 3.
Minor points:
1. Reformat Table 1 by making the table wider or the font-size smaller so that the words do not touch the frame of the table.
2. In 4.4. hnRNP K - mRNA localization, transcription, translation and splicing, three paragraphs are related with role of hnRNP K in lung cancer, rectal adenocarcinoma and melanoma respectively. Please focus on hematological malignancies and avoid discussing too much about solid tumors although they have been extensively studied previously.
3. In Paragraph 4 of Section 2. RNA-binding proteins, please add reference to Commonly, RBPs contain multiple repeats of the same RBD, which contributes to an enhanced specificity and affinity for their target mRNAs.
4. In Paragraph 2 of Section 4. RNA-binding proteins in hematological malignancies, please add reference to RBP modifications result in biological changes such as altered hematopoietic lineage development or abnormalities that, in many circumstances, can result in bone marrow failure.
Author Response
We thank the reviewer for all the comments. We upload a word document with all the detailed responses to those suggestions and comments.

Reviewer 2 Report
The authors report the role of RNA-binding proteins in hematological malignancies.
1. The authors should provide the schema of RNA binding proteins (e.g., structure, signaling) in a Figure. It will be benefit for the reader.
2. The authors should provide the clinical trial information and describe in the text.
Author Response

(The authors gave the same response as above.)

Round 2
Reviewer 2 Report
none